# Cystatin C, a Controversial Biomarker in Hypothyroid Patients under Levothyroxine Therapy: THYRenal, a Pilot Cohort Observational Study

**DOI:** 10.3390/jcm9092958

**Published:** 2020-09-13

**Authors:** Marta Greco, Daniela Patrizia Foti, Antonio Aversa, Giorgio Fuiano, Antonio Brunetti, Mariadelina Simeoni

**Affiliations:** 1Department of Health Sciences, University “Magna Græcia” of Catanzaro, 88100 Catanzaro, Italy; marta.greco@unicz.it (M.G.); foti@unicz.it (D.P.F.); 2Department of Experimental and Clinical Medicine, University “Magna Græcia” of Catanzaro, 88100 Catanzaro, Italy; aversa@unicz.it; 3Department of Medical and Surgical Sciences, University “Magna Græcia” of Catanzaro, 88100 Catanzaro, Italy; fuiano@unicz.it

**Keywords:** renal function, thyroid disease, hypothyroidism, Cystatin C, eGFR, mGFR, Levothyroxine

## Abstract

Background: Cystatin C (Cys-C) is recognized as one of the most reliable renal function parameters in the general population, although it might be biased by thyroid status. Herein, we tested Cys-C and conventional renal parameters in a cohort of hypothyroid patients treated with Levothyroxine. Methods: Eighty-four hypothyroid patients were recruited and subgrouped according to their serum thyroid-stimulating hormone (TSH) values as a paradigm for therapeutic targeting (*n* = 54, optimal TSH range = 0.5–2 µIU/mL; *n* = 30, TSH > 2µIU/mL). Serum Cys-C, creatinine, measured and estimated glomerular filtration rates (mGFR and eGFR) were assessed. Results—mGFR and eGFR were comparable among the two subgroups, whereas Cys-C was significantly higher in patients with suboptimal TSH values (>2 µIU/mL) (*p* < 0.0001). TSH significantly correlated with Cys-C in the overall patient group, and in the subgroup with TSH above the target value (>2 µIU/mL). Out of 20 patients with abnormal Cys-C, 19 had suboptimal TSH levels. Receiver operating characteristic (ROC) analysis indicated Cys-C as a moderately accurate diagnostic tool (AUC = 0.871) to assess Levothyroxine replacement efficacy in hypothyroid patients (63% sensitivity, and 98% specificity). Conclusions: The observation of increased serum Cys-C in patients with suboptimal TSH would suggest the importance of a careful interpretation by clinicians of this biomarker in the case of hypothyroid patients.

## 1. Introduction

Hypothyroidism is a major health issue associated with several organ complications, including kidney disease. The relationship between kidney and thyroid has been extensively studied and the influence of thyroid hormones on kidney development and on renal structure and function is well-established [1,2,3,4,5]. Thyroid hormones affect renal mass, renal blood flow, glomerular filtration rate (GFR), tubular electrolyte and acid-base transports, water excretion capacity, adrenergic receptors, as well as the renin-angiotensin-aldosterone system (RAAS), as demonstrated in both humans and animal models [4,5,6,7,8,9,10,11]. In several studies, the association of overt and/or subclinical primary hypothyroidism with GFR reduction and elevated serum creatinine has been demonstrated, in addition to effects on salt and water homeostasis [4,8,12]. These effects are generally normalized following levothyroxine therapy in adult hypothyroid patients and can be improved in hypothyroid patients with chronic kidney disease (CKD) [4,13,14,15]. In addition, it is remarkable that patients with appropriately treated vs. untreated thyroid disorders have a lower risk of developing renal dysfunctions [1,4,16,17].

In light of the importance of an early detection of renal damage in hypothyroidism, a correct therapeutic approach is fundamental to decrease the risk of CKD development [13,14,15]. Similarly, a precise and reliable measure of renal function in hypothyroid patients is strongly desirable. Both aspects are still open issues [15,18,19,20,21].

Levothyroxine (L-T4) is the standard of care in hypothyroidism. In the most common primary hypothyroidism, L-T4 daily dose has to be tailored to the patient on the basis of serum thyroid-stimulating hormone (TSH) levels [22]. In fact, according to the American Thyroid Association and the American Association of Clinical Endocrinologists guidelines, TSH is considered the best marker of thyroid dysfunction [23]. Although it has been accepted that the TSH reference range differs with age and should be between 0.4 and 4.5 µIU/mL in adults in relation to treatment, in primary hypothyroidism, a serum TSH value between 0.5 and 2.0 µIU/mL is the therapeutic target for a standard L-T4 replacement dose [24]. This goal is, however, not reached in at least 30% of patients due to inconsistent adherence to therapy [25].

On the other hand, the use of conventional, creatinine-derived renal function parameters in hypothyroidism are influenced by several conditions, including sex, age, muscle mass, inflammation, and uremia, and may lead to inaccurate results. Cystatin C (Cys-C) is a cysteine protease inhibitor constantly synthesized by nucleated cells, filtered through the glomerulus, and reabsorbed in the proximal tubules [26,27]. Compared to traditional parameters, Cys-C is considered a more reliable renal function marker in the assessment of GFR in the general population. In certain circumstances, including CKD, this marker shows higher sensitivity and specificity than traditional creatinine-derived parameters, and a more evident association with clinical outcomes [28,29]. This index is, however, influenced by cell turnover, so that its potential diagnostic use in the renal function assessment in patients affected by thyroid dysfunction has to be established [30]. Although some studies have already been carried out to evaluate Cys-C in hypothyroidism, as well as in hyperthyroidism [18,19,20,21,31], no reports have investigated how L-T4 replacement therapy in hypothyroid patients could impact on serum Cys-C compared to renal conventional markers. Thus, this pilot observational study was conducted to establish the relationship between thyroid and renal function in hypothyroid patients under treatment with L-T4 using both creatinine-derived parameters and Cys-C.

## 2. Materials and Methods

Eighty-four patients, referred to the Unit of Endocrinology at the University Hospital of Catanzaro (Italy), were selected from a stable outpatient population. Patients were affected by hypothyroidism on hormone replacement therapy with L-T4 and all of them provided the consent to participate in this research study. The study was conducted in accordance with the 1964 Helsinki Declaration and approved by the Ethics Committee “Regione Calabria, Sezione Area Centro” (Protocol n. 124, 14.5.2015). All participants gave written informed consent.

To be enrolled, patients had to meet the following criteria: age ranging 30–65 years, Caucasian race, unmodified L-T4 replacement therapy for at least three months and TSH results between 0.5 and 5 μIU/mL, BMI < 35 kg/m^2^, and no abnormalities in standard urine test. Patients affected by cancer, diabetes mellitus, arterial hypertension, glomerulopathies, tubulopathies, and/or autoimmune systemic diseases were excluded. In addition, patients on hemodialysis or peritoneal dialysis and kidney transplant recipients could not be enrolled. Furthermore, patients with a history of symptomatic hypothyroidism for more than four months before L-T4 treatment initiation were excluded, as well as patients treated with either non-steroidal and/or steroidal anti-inflammatory drugs. On the whole, criteria of eligibility aimed to restrict conditions, including aging and/or pathologies different from the thyroid status, which could potentially be a threat for renal function, or intrinsically alter Cys-C levels, such as corticosteroids. For each recruited patient, medical history and demographic data were collected. A careful physical examination was also performed. In order to assess thyroid and renal markers, all patients had blood samples taken for fasting lab tests and had to provide morning and 24 h urine samples. The following lab tests were finally performed: ultrasensitive TSH by an immunochemiluminescent test (Siemens Healthcare Diagnostics, Marburg, Germany); estimated GFR (eGFR) by using the CKD Epidemiology Collaboration (CKD-EPI) formula; measured GFR (mGFR) calculated on serum and 24 h urinary creatinine assessed by a kinetic enzymatic method (Roche Diagnostics, Risch-Rotkreuz, Switzerland); Cys-C assessed by a nephelometric method (Siemens Healthcare Diagnostics, Marburg, Germany). Renal function was considered impaired when eGFR and mGFR were <90 mL/min/1.73 m^2^ [32], serum creatinine was >0.9 mg/dL in females and >1.2 mg/dL in males [33], and Cys-C was ≥ 0.99 mg/L. The TSH value used as the cutoff of therapeutic efficacy was 2.0 µIU/mL.

### Statistical Analysis

Statistical analysis was conducted with SPSS 20.0 for Windows. The normal distribution of variables in the population study has been expressed as mean values ± standard deviation. Data were analyzed using *t*-test for independent parameters and Pearson’s correlation test with linear regression. The sample size in an adaptative design has been calculated by NCSS Pass 20, while a posteriori power analysis (99.9%) has been determined by G*Power.

Receiver operating characteristic (ROC) analysis was performed to assess sensitivity and specificity of Cys-C in identifying treated hypothyroid patients in which serum TSH was above the target value (>2.0 µIU/mL). Statistical significance was set at a *p* value < 0.05.

## 3. Results

### 3.1. Comparison between L-T4 Treated Hypothyroid Patients Reaching Optimal vs. Suboptimal TSH Therapeutic Targets

Demographic and clinical characteristics of the total enrolled population are shown in Table 1.

Of note, most patients are females, and this observation reflects the higher prevalence of thyroid dysfunctions in adult women. In 57 patients out of 84 (68%), hypothyroidism was due to autoimmune chronic thyroiditis, as proved by the positivity rates of anti-thyroid peroxidase and anti-thyroglobulin autoantibodies; in 21 patients (25%), hypothyroidism was caused by thyroidectomy for benign multinodular goiter; and in six cases (7%), it was due to other causes, mainly antibody-negative autoimmune thyroiditis. Blood pressure, as well as renal function parameters [32,33], were within the normal range.

Results from the same cohort were then reanalyzed according to the values of TSH reached during L-T4 replacement therapy. Optimal TSH results were considered between 0.5 and 2.0 μIU/mL, while TSH > 2 μIU/mL was considered suboptimal (Table 2).

Patients in the two groups were comparable for age and BMI. Differences in blood pressure could be attributed to the proportionally higher number of menopausal women in the group with TSH in the optimal range. As expected, the two cohorts were also comparable for FT3 and FT4 levels (Table 2).

In relation to the renal functional parameters, no differences were found in serum creatinine levels, mGFR, and CKD-EPI eGFR between the two groups, whereas serum Cys-C was found to be significantly higher in patients with suboptimal TSH levels (Table 2).

### 3.2. Correlation Analyses between Renal Functional Parameters

As reported in Figure 1, no correlation between Cys-C and other renal functional parameters was found in the total study population. Moreover, serum levels of Cys-C were not found to be significantly correlated with gender and age, in line with data from the literature.

### 3.3. Correlation Analyses between TSH and Renal Functional Parameters

We then studied the correlation between TSH as a marker of L-T4 therapeutic efficacy and the conventional renal parameters under investigation. In the overall population, TSH did not correlate with creatinine, mGFR, and CKD-EPI eGFR, (Figure 2a–c). Conversely, TSH and Cys-C showed a highly significant positive correlation (*p* < 0.0001; Figure 2d).

This last result was clearly linked to the population with suboptimal TSH values. In fact, despite the small sample size, in the subgroup of patients with TSH > 2 µIU/mL, a significant correlation between Cys-C and TSH was observed (*p* = 0.010; Figure 3a), while in the subgroup of patients with TSH ≤ 2 µIU/mL this correlation failed to be significant (*p* = 0.095; Figure 3b).

### 3.4. Statistical Measures Linking Serum Cys-C to Medical Diagnosis of Hypothyroid Patients with Suboptimal TSH Target

In the overall population, 20 patients showed Cys-C serum levels above the reference range: 19 were in the subgroup with TSH above the target value (>2 µIU/mL), while one was part of the subgroup with optimal TSH (≤2 µIU/mL), assessing a Cys-C specificity of 98%. On the other hand, among the 30 patients with suboptimal TSH, 11 had Cys-C serum levels in the reference range, so that, in the context herein described, the diagnostic sensitivity of Cys-C remained low (63%).

As a measurement of diagnostic accuracy of the thyroid status, we performed ROC curve analysis. The areas under the curves (AUC) of serum Cys-C levels, creatinine, mGFR, and CKD-EPI eGFR were 0.871, 0.548, 0.596, and 0.613, respectively (Figure 4). Therefore, among the different kidney indexes examined, Cys-C seemed to be the renal functional parameter that might help identify patients with hypothyroidism under suboptimal treatment response.

## 4. Discussion

In this study, we investigated the relationship between renal and thyroid function in an outpatient hypothyroid population, in which treatment with L-T4 had apparently restored euthyroidism. Renal function was explored with a multi-parametric approach to compare creatinine-derived renal function parameters and Cys-C. As widely reported in the literature, hypothyroidism can influence renal function by inducing both hemodynamic and non-hemodynamic changes in the kidney. In this regard, undoubtedly, the duration of untreated thyroid deficiency could lead, mainly via RAAS dysregulation, to an irreversible renal damage with possible CKD development [8,9]. We therefore excluded, from our study, patients with a history of symptomatic hypothyroidism for more than 4 months, and enrolled patients with a stable L-T4 replacement therapy and TSH levels < 5 μIU/mL. We hypothesized that these selection criteria might have positively influenced renal functional parameters, as data on renal function before L-T4 initiation were not available.

In our case, when creatinine-derived parameters in the whole study population were considered, kidney function seemed preserved. However, we found higher levels of Cys-C in patients with TSH values above the optimal range. In addition, in the overall assessment, CKD-EPI eGFR, mGFR, and serum creatinine did not, whereas Cys-C definitely showed a positive correlation with serum TSH.

Some reports have investigated Cys-C levels in different conditions of thyroid dysfunctions. In the case of overt hyperthyroidism and hypothyroidism, Cys-C levels were found to be increased and decreased, respectively [19,31], while the restoration of euthyroidism normalized mean serum Cys-C. Similar and parallel results have emerged from subclinical conditions of thyroid disorders [21]. Conversely, a more recent study in a male Chinese population of more than 8000 patients showed a significant association of increased levels of Cys-C with both subclinical hypo- and hyperthyroidism [34]. While these authors concluded that Cys-C should not be used in patients with thyroid dysfunctions, interestingly, they also showed a higher prevalence of CKD on the basis of Cys-C-derived eGFR for TSH values up to 7 μIU/mL, and a higher prevalence of CKD on the basis of creatinine-derived eGFR for TSH values over 7 μIU/mL.

Our investigation, for the first time, tried to disentangle how Cys-C related with patients with acquired primary hypothyroidism in current euthyroid state following a stable L-T4 replacement therapy. We believe that the finding of higher Cys-C levels in the subset of patients with suboptimal TSH levels is quite relevant and opens new interesting and opposite scenarios. Is Cys-C more accurate than conventional renal function parameters in detecting early renal damage also in patients with treated hypothyroidism? Or, conversely, is Cys-C an unreliable renal function marker in this specific patient typology, and, instead, could rather reflect thyroid hormone biological activity?

As for the first hypothesis, Cys-C is increasingly accepted as a more sensitive renal function marker than serum creatinine to detect mild reductions of GFR in several renal diseases [28,29,35]. Unlike creatinine, Cys-C has been shown to be independent from gender, age, height, and body composition and to be superior to serum creatinine in specific cases (children, adolescents, elderly, pregnant women) [36,37]. Being unaffected by muscle mass, Cys-C has proved to be preferable to creatinine in patients with sarcopenia or abnormal creatinine excretion. In addition, Cys-C estimates the decline in renal function more precociously than conventional markers in a series of diseases, including diabetes mellitus [38,39]. It can therefore be hypothesized that in our population, Cys-C might have performed better than mGFR and eGFR in predicting an early renal damage onset, possibly due to under-controlled hypothyroidism. Although control subjects and/or patients with impaired renal function are required for verification, this hypothesis cannot be discarded. In coherence with this assumption, in euglycemic euthyroid patients, TSH levels independently associate with renal function and CKD [16,40]. On the other hand, in our study, Cys-C appears to be strongly correlated with TSH levels, and noticeably, 19 out of 20 patients with above-range Cys-C levels were among the subgroup with a suboptimal TSH target, so that even the second hypothesis becomes feasible. This intriguing interpretation suggests Cys-C as an objective marker of thyroid hormone biological signaling, as described for LDL and total cholesterol, as well as for sex hormone-binding globulin [24]. Since mild thyroid failure cannot be caught by thyroid hormone levels [41], speculatively, altered levels of Cys-C could represent a warning for L-T4 dose adjustment in hypothyroid patients. In our study, the diagnostic sensitivity of Cys-C towards patients with suboptimal TSH, however, was low (63%). This observation, consistent with a previous report [19], seems to limit the use of Cys-C as a marker of peripheral thyroid hormone action.

As a pilot study, the main limitations are related to the small size of the analyzed population and the lack of untreated euthyroid and hypothyroid controls. As such, the hypothesized concept needs to be first addressed before launching into a larger study that would require appropriate control cohorts. However, we believe that the homogeneity of the thyroid disorder and estimated exposure time to untreated disease, as well as the enrollment of patients within a narrow age range, which excluded the elderly, and the inclusion of cases with TSH levels within or close to the reference range might add relevance and reliability to our results.

Although further studies are necessary to demonstrate the appropriate role of Cys-C in treated hypothyroid patients, the finding of increased serum Cys-C in cases with suboptimal TSH suggests the importance of correcting L-T4 dosing to prevent early renal impairment.

## 5. Conclusions

The evaluation of correct renal function in patients with thyroid disorders represents an important issue, as hypothyroidism is known to increase the risk of CKD.

Cys-C may be often found increased in hypothyroid patients with suboptimal L-T4 replacement therapy, representing a warning for physicians to check thyroid function. In these cases, with respect to the current creatinine-based measures, Cys-C may represent a more sensitive index of early renal impairment, but it also has the potential to be a biomarker of inadequate hormonal management in hypothyroidism that, if not corrected, puts the patient at risk of CKD.

As a pilot study, further investigations are desirable to better understand the role and the use of Cys-C in hypothyroidism.

## Figures and Tables

**Figure 1 jcm-09-02958-f001:**
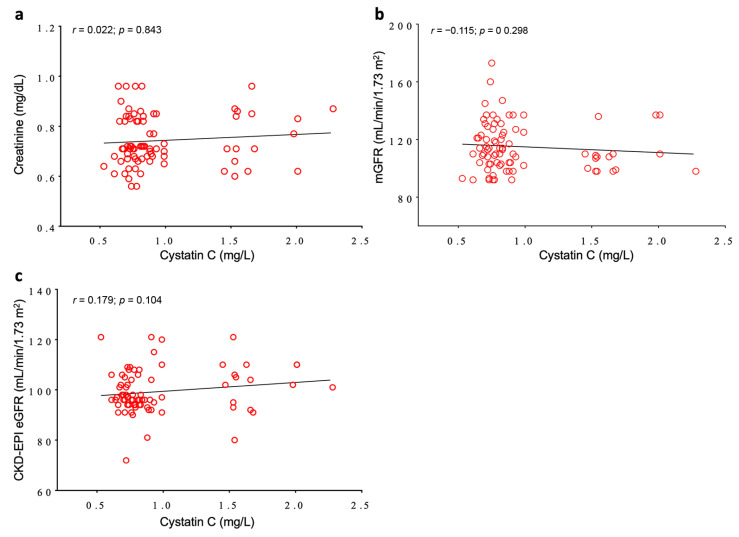
Pearson’s correlation analysis between Cystatin C and other renal functional parameters in the total study population. (**a**) Cystatin C (Cys-C) vs. creatinine; (**b**) Cys-C vs. mGFR; (**c**) Cys-C vs. CKD-EPI eGFR.

**Figure 2 jcm-09-02958-f002:**
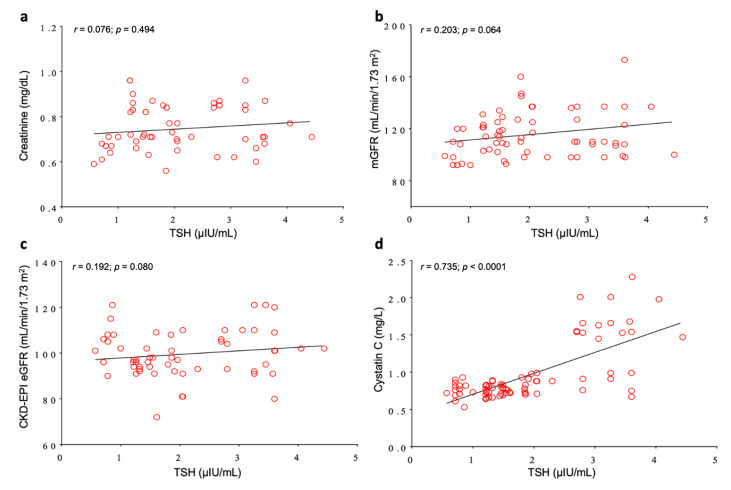
Pearson’s correlation analysis between TSH and renal functional parameters in the total study population. (**a**) TSH vs. creatinine; (**b**) TSH vs. mGFR; (**c**) TSH vs. CKD-EPI eGFR; (**d**) TSH vs. Cystatin C.

**Figure 3 jcm-09-02958-f003:**
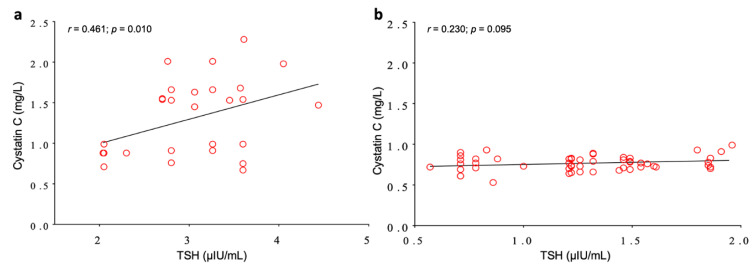
Pearson’s correlation analysis between TSH and Cystatin C in the subgroup with TSH values > 2 µIU/mL (**a**), and in the subgroup with TSH values ≤ 2 µIU/mL (**b**).

**Figure 4 jcm-09-02958-f004:**
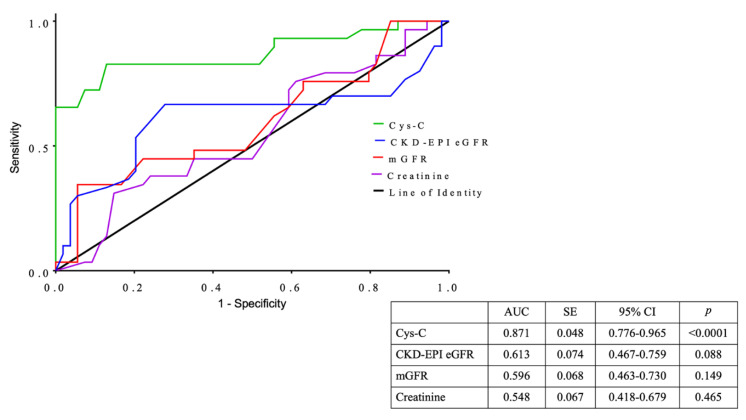
Receiver operating characteristic (ROC) curves of renal functional parameters for prediction of L-T4 treated hypothyroid patients with suboptimal TSH values (>2.0 µIU/mL). AUC = area under curve; SE = standard error; CI = confidence interval.

**Table 1 jcm-09-02958-t001:** General characteristics and laboratory parameters of the study cohort.

	Study Cohort
Race	Caucasian
*N*	84 (56 ♀–28 ♂)
Age (yrs)	53 ± 8.5
BMI (kg/m^2^)	28.76 ± 3.3
Systolic blood pressure (mm Hg)	123 ± 10.2
Diastolic blood Pressure (mm Hg)	75 ± 8.8
TSH (μIU/mL)	1.9 ± 1.0
FT3 (pg/mL)	3.23 ± 0.3
FT4 (ng/dL)	1.33 ± 0.2
Creatinine (mg/dL)	0.74 ± 0.1
mGFR (mL/min/1.73 m^2^)	115 ± 16.6
CKD-EPI eGFR (mL/min/1.73 m^2^)	99 ± 8.8
Cystatin C (mg/L)	0.97 ± 0.4

BMI: body mass index; eGFR: estimated glomerular filtration rate; mGFR: measured glomerular filtration rate. Data are expressed as mean values ± standard deviation.

**Table 2 jcm-09-02958-t002:** Characteristics of the study cohort in relation to serum thyroid-stimulating hormone (TSH) levels.

	Patients with Optimal TSH	Patients with Suboptimal TSH	*p*
*N*	54 (36 ♀–18 ♂)	30 (20 ♀–10 ♂)	
Age (yrs)	53 ± 8.5	51 ± 8.4	0.208
BMI (kg/m^2^)	28.29 ± 3.1	29.59 ± 3.5	0.087
Systolic blood pressure (mm Hg)	127 ± 9.7	117 ± 8.4	<0.0001
Diastolic blood Pressure (mm Hg)	75 ± 8.3	72 ± 7.5	0.038
TSH (μIU/mL)	1.28 ± 0.4	3.10 ± 0.6	<0.0001
FT3 (pg/mL)	3.24 ± 0.3	3.22 ± 0.4	0.821
FT4 (ng/dL)	1.32 ± 0.2	1.34 ± 0.2	0.716
Creatinine (mg/dL)	0.74 ± 0.1	0.75 ± 0.1	0.730
mGFR (mL/min/1.73 m^2^)	113 ± 15.2	119 ± 18.6	0.125
CKD-EPI eGFR (mL/min/1.73 m^2^)	98 ± 7.2	102 ± 11.03	0.076
Cystatin C (mg/L)	0.77 ± 0.1	1.34 ± 0.5	<0.0001

TSH levels between 0.5 and 2.0 μIU/mL are considered optimal [24]. BMI: body mass index; eGFR: estimated glomerular filtration rate; mGFR: measured glomerular filtration rate. Data are expressed as mean values ± standard deviation. *p* < 0.05 is considered significant (bold).

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
