# Peer review of "Cystatin C, a Controversial Biomarker in Hypothyroid Patients under Levothyroxine Therapy: THYRenal, a Pilot Cohort Observational Study"

_jcm, 2020, doi:10.3390/jcm9092958_

Round 1

Reviewer 1 Report

Dear authors,

your paper highliths the importance of adequate LT4 substitution for renal function in hypothyroidism, emphasiying the inadequcy of TSH measurement alone.

The methodology did not state whether subjects had normal blood pressure with antihypertensive therapy or did not suffer from hypertension.

In exclusion criteria you did not mention previous use of steroids that may affect cystatin findings.

In view of the importance of determining cystatin as an early marker of renal impairment, although there are not many studies on this topic, the discussion could be more extensive.

Kind regards,

reviewer

Author Response

Authors: A point-by-point response to the reviewer 1 is provided in blue text.

Authors: We thank the reviewer for his/her time and constructive comments, that have allowed us to improve this manuscript.

The methodology did not state whether subjects had normal blood pressure with antihypertensive therapy or did not suffer from hypertension.

Authors: Patients suffering from arterial hypertension were not enrolled in this study. This is now clearer reported (line 91).

In exclusion criteria you did not mention previous use of steroids that may affect cystatin findings.

Authors: Patients under treatment with both non-steroidal and/or steroidal anti-inflammatory drugs were excluded. This  has been now added in the “Materials and Methods” (lines 94-95).

In view of the importance of determining cystatin as an early marker of renal impairment, although there are not many studies on this topic, the discussion could be more extensive.

Authors: To accomplish the reviewer’s request, in addition to what had been already mentioned in the Introduction (lines 68-71), we now extended the discussion and related bibliography in relation to Cys-C as an early marker of renal impairment (lines 223-229).

Reviewer 2 Report

Cystatin C, A Controversial Biomarker in Hypothyroid Patients under Levothyroxine Therapy: THYRenal, A Pilot Cohort Observational Study

The study is really well written and present an interesting concept of Cys-C as a marker of association of thyroid treatment with renal function. Some minor points need to be clarified

Line 82 - age range from 30-65 years – can you explain the reason for this and other criteria of inclusion to the study?

Overtargeted TSH – when I was reading the article this part was a bit confusing to me, please explain what exact value of TSH means overtargeted. Maybe it would be enough to put this (or paraphrased) fragment: “TSH was above the target value (> 2.0 μIU/mL)” somewhere at the beginning of the article

Author Response

Authors: A point-by-point response to the reviewer 2 is provided in blue text.

Authors: We thank the reviewer for his/her time and constructive comments, that have allowed us to improve this manuscript.

Line 82 - age range from 30-65 years – can you explain the reason for this and other criteria of inclusion to the study?

Authors: We have chosen to enroll an adult study population (30-65 years), that excluded the elderly to limit potential co-morbidities linked to aging.  We also excluded any known renal disorder and risk factors for CKD, such as diabetes mellitus and arterial hypertension, as well as other pathological conditions that may represent a potential, independent risk for CKD in our population, such as systemic autoimmune diseases.  Also, chronic therapies other that Levothyroxine, including non-steroidal and/or steroidal anti-inflammatory drugs were excluded, as the latter may influence cystatin C levels.   Ideally, we aimed to have a quite selected study population, with little if any co-morbidities, to better focus on the role of  Levothyroxine treatment in renal function. This is now stated (lines 95-98).

Overtargeted TSH – when I was reading the article this part was a bit confusing to me, please explain what exact value of TSH means overtargeted. Maybe it would be enough to put this (or paraphrased) fragment: “TSH was above the target value (2.0 μIU/mL)” somewhere at the beginning of the article.

AuthorsAs suggested, to improve clarity, we corrected the expression “over-target TSH” throughout the text with either “suboptimal”, already defined in the abstract (line 27), or  with the locution “TSH above the target value (> 2.0 μIU/mL)”.